# Antiproliferative and Pro-Apoptotic Effects of a Phenolic-Rich Extract from *Lycium barbarum* Fruits on Human Papillomavirus (HPV) 16-Positive Head Cancer Cell Lines

**DOI:** 10.3390/molecules27113568

**Published:** 2022-06-02

**Authors:** Alberto Peraza-Labrador, Diana Marcela Buitrago, Ericsson Coy-Barrera, Sandra J. Perdomo-Lara

**Affiliations:** 1Unit of Basic Oral Investigation-UIBO, School of Dentistry, Universidad El Bosque, Bogotá 110121, Colombia; aperaza@unbosque.edu.co (A.P.-L.); buitragodianam@unbosque.edu.co (D.M.B.); 2Cellular and Molecular Immunology Group-INMUBO, School of Dentistry, Universidad El Bosque, Bogotá 110121, Colombia; 3Bioorganic Chemistry Laboratory, Department of Chemistry, Universidad Militar Nueva Granada, Cajicá 250247, Colombia; ericsson.coy@unimilitar.edu.co

**Keywords:** head and neck cancer, human papillomavirus, *Lycium barbarum*, antiproliferative, pro-apoptotic activity

## Abstract

The in vitro antiproliferative activity of a phenolic-rich extract from *Lycium barbarum* fruits against head and neck HPV16 squamous cell carcinoma (OSCC) has been demonstrated, indicating for the first time that *L. barbarum* extract inhibits human papillomavirus (HPV) type 16 cell lines. Ethanol extract of *L. barbarum* was used for cell viability evaluation on SCC090, CAL27, and HGnF cell lines. After 24 and 48 h, the cell cycle effect of *L. barbarum* extract (at 1.0, 10, and 100 µg/mL) was measured via flow cytometry. In addition, the mRNA expression on E6/E7 and p53 via RT-PCR and the expression of p16, p53, Ki-67, and Bcl-2 via immunohistochemistry were also determined. Untreated cells, 20 µM cisplatin, and a *Camellia sinensis*-derived extract were used as negative and positive controls, respectively. We demonstrated that the studied *L. barbarum* extract resulted in G_0_/G_1_ arrest and S phase accumulation in SCC090 at 1.0 and 10 μg/mL. A reduction in mRNA levels of E6/E7 oncogenes (*p* < 0.05) with p53 overexpression was also observed through PCR, while immunohistochemical analyses indicated p16 overexpression (*p* > 0.05) and a decrease in p53 overexpression. The observed effects were associated with anticancer and immunomodulatory phenolics, such as flavonols/flavan-3-ols and tyramine-conjugated hydroxycinnamic acid amides, identified in the studied extract. These findings revealed that the phenolic-rich extract of *L. barbarum* fruits has promising properties to be considered further for developing new therapies against oral and oropharyngeal HPV lesions.

## 1. Introduction

The occurrence of oral and oropharyngeal squamous cell carcinoma (OSCC) has rapidly increased over the past several decades and is a global public health problem. According to the Global Cancer Observatory (GCO), OSCC is responsible for 0.5% (92,887) of all new cancer cases diagnosed annually and 0.5% (51,005) of all cancer deaths [1]. The increased incidence of oral cancer in younger patients without alcohol and smoking history is associated with oral sexual behavior and high-risk human papillomavirus (Hr-HPV) infection [2,3]. According to the World Health Organization (WHO), the HPV 16 Hr-genotype has been implicated in 84% of OSCC [4,5]. Although most HPV infections are eliminated naturally, the persistence of Hr-HPV infection is the major risk factor for the development of premalignant lesions and OSCC [6].

Despite a vaccine having promising results in preventing HPV-associated cervical cancer, its effectiveness in the oral and oropharyngeal areas has not yet been demonstrated. The Centers for Disease Control and Prevention in the United States have promoted vaccination in children and teenagers [7], and some studies have found the vaccine effective on head and neck cancer in the short term [2,8], but the long-term effects of vaccine therapies will not be observed until the year 2040 [9]. In this context, surgery and chemotherapy are always the first strategies for treating a patient with an advanced OSCC. However, the use of some chemotherapeutic drugs has been limited by the lack of selectivity and specificity against HPV, as well as numerous side effects [10,11]. Thus, there is a clinical need to develop alternative therapeutic strategies to manage HPV-positive patients.

Since ancient times, plants and fungi have been used as medicinal products against different pathologies and, more recently, in preparations/elaborations of active metabolite mixtures to produce new active principles. These explorations have played a crucial role in modern drug discovery [12], even in the 21st century [13], looking for alternative therapeutics to be used against emergent diseases such as OSCC. Furthermore, natural products derived from medicinal plants with a well-defined safety and efficacy profile are more accessible as treatments and easily transferred to clinical practice [14]. These features can be rationalized since hundreds of interrelated chemical compounds are present in living or dried plant-derived ingredients. These chemical entities can generally exhibit diverse biological and therapeutic effects, even if the entire plant preparation is employed as a phytotherapeutic [15]. Hence, bioactive compounds obtained from low-toxicity natural resources offer new options for developing effective chemotherapeutic or adjuvant therapy for cancer treatment [16], since natural products are generally less toxic than synthetic products [17].

Among botanical sources of bioactives with evidenced interesting biological/pharmacological profiles is *Lycium barbarum*, a solanaceous, deciduous woody perennial (commonly called Chinese wolfberry or Goji) exploited as a functional food and traditional herbal medicine in China since ancient times and therefore registered in the *Chinese Pharmacopoeia* [18]. Its phytoconstituents from seeds, leaves, and especially fruits have received particular attention in the past two decades because of their explored in vitro and in vivo properties: antioxidant, antihypertensive, antihyperglycemic, antitumor, antihyperlipidemic, and anti-Alzheimer, among others [19]. Polysaccharides (usually present in water-soluble extracts of the berries) and flavonoids, phenolics, and carotenoids (present in organic solvent-soluble extracts such as ethanol) are considered the active principles for the reported biological activity of *L. barbarum* [20,21]. Indeed, some studies have shown the promising effects of ethanolic extracts of this plant on several cancer cell lines [22], so we hypothesized its plausible effectiveness on OSCC, particularly the fruit-derived extract containing active phenolics [23,24].

Hence, the present study describes the molecular mechanism underlying the antiproliferative potential of a phenolic-rich extract of *L. barbarum* fruit on a naturally transformed HPV-16 positive SCC090 cell line. Our data suggest that *L. barbarum* inhibits the growth of tongue cell line SCC090 by inducing G_1_/S cell cycle arrest and the overexpression of p53 and p16 proteins. Interestingly, *L. barbarum* significantly reduced the expression of viral oncoproteins E6 and E7, which further suggests the therapeutic potential of this natural extract in OSCC.

## 2. Results

### 2.1. Chemical Characterization of L. Barbarum Extract

Extract of *L. barbarum* fruits was firstly characterized by total flavonoid and phenolic content (i.e., TPC and TFC, respectively) measurements. The TPC of the test extract was found to be 46.31 ± 1.07 mg GAE/g Edw; TFC was 26.53 ± 0.98 mg CE/g Edw. Liquid chromatography coupled with high-resolution mass spectrometry with electrospray ionization (LC-ESI-HRMS) analysis gave additional information on the chemical composition of test *L. barbarum* extract beyond TPC and TFC. Figure 1 presents the total ion chromatogram (TIC). Compounds were identified based on detailed scrutiny of their mass spectra using the precursor ion, fragment ions, and comparison of the fragmentation patterns with molecules described in the literature [23,25,26,27]. Table 1 summarizes the identification of these compounds, and they are numbered according to their retention times. Relative abundance (%) was also calculated based on the normalized peak intensity from the TIC. Fifteen main compounds were detected and identified, including four flavonoid glucosides (combined relative abundance (JRA) = 49.60%), a conjugated phenylpropanoid (JRA = 21.03), four free flavonoids (JRA = 9.05%), and four nitrogen-containing compounds, i.e., five hydroxycinnamic acid amides (HCAA) (JRA = 17.45%), and a lignamide (JRA = 2.87%).

The most abundant compound (34.87%) in the test extract was 3′,7-O-dimethylcatechin 3-O-glucoside (peak 6, retention time = 7.92 min, C_23_H_28_O_11_, error = −1.66) (Table 1 and Figure 1), a flavan-3-ol glycoside-type compound, deduced from the fragment *m*/*z* 319.1172 that corresponds to the aglycone substructure (O-dimethylcatechin). Methyl group at C7 in ring A was inferred by the fragment ion *m*/*z* 197.0802, the specific fragment for the 7-methoxychromane-3,5-diol moiety. In this regard, ring B comprises the 3′-methoxyphenol moiety when the biosynthetic route via flavonoid 3′,5′-hydroxylase is activated in berries [28]. A common fragment was observed in the case of amides, corresponding to the tyramine- and dopamine-related residues, which were relevant for identification purposes [29]. Identical high-resolution mass spectra-derived fragments and accurate mass analyses were performed to identify the other detected compounds in the studied extract, listed in Table 1. The TIC separated the compounds into two particular zones related to flavonoids (free and conjugated) and amides. Flavonoid aglycones were related to flavan-3-ols (e.g., catechin) and flavonols (e.g., quercetin and luteolin) already described for Lycium plants [23]. A series of five HCAAs and a dihydrobenzofuran neolignan conjugated with tyramine and (methyl)dopamine were also detected, involving caffeoyl, coumaroyl, and feruloyl moieties [30].

### 2.2. Antiproliferative Activity and SI of L. barbarum Extract

The effect of the ethanol extracts of *L. barbarum* and *C. sinensis* against three cell lines, i.e., SCC090 (HPV16-positive squamous cell carcinoma (SCC) from tongue), CAL27 (HPV16-negative SCC from tongue), and HGnF (normal human gingival fibroblasts), was evaluated at concentrations between 10 and 2000 μg/mL using the Alamar blue assay. The chemotherapeutic drug cisplatin, used as a control, exhibited a half-maximal inhibitory concentration (IC_50_) of 19.49 μM, equivalent to 5.84 μg/mL over the SCC090 cell line (Table 2), consistent with those already described by other authors [31]. Both studied extracts exhibited modest antiproliferative effects on cancer cells and normal fibroblasts at the concentrations and times evaluated (IC_50_ > 100 μg/mL) (Figure 2).

However, despite the modest activity, IC_50_ values differed between extracts and cell lines. This observation, combined with reported data on anticancer and immunomodulatory effects of *L. barbarum* fruit [22], gained our attention (Table 2). CAL27 was the most susceptible cell line (IC_50_ = 266.2 µg/mL), and the normal fibroblasts were the least susceptible cell line (IC_50_ = 626.5 µg/mL). A similar performance was observed for *C. sinensis* extract. In this sense, tested extracts exhibited better cytotoxic activity against cancer cell lines, involving selectivity indices (SIs) in the 1.3–2.4 range. This fact was taken as the starting point to subsequent experiments, since the selectivity of chemopreventive agents means that cancer cells are more susceptible [32], less toxic to normal cells, and have the most negligible effect on them [32]. *L. barbarum* extract exhibited selectivity for CAL27 and SCC090 (SI > 1.4) (Table 2) and lesser toxicity for HGnF cells. Hence, the safety of the *L. barbarum* extract against normal HGnF cells and the selectivity against cancer cells were confirmed. Based on these results, we chose non-cytotoxic concentrations of *L. barbarum* (i.e., 1.0, 10, and 100 µg/mL) in our subsequent assays.

### 2.3. L. barbarum Induces G1/S Phase Arrest in SCC090 Cell Lines in a Dose-Dependent Manner

We analyzed the cell cycle distribution in SCC090 and CAL27 cell lines at 48 h to determine the mechanism behind *L. barbarum*-mediated regulation of growth in OSCC cancer cells. Flow cytometry analysis (Figure 3) showed that *L. barbarum* at 1.0 µg/mL exhibited an increase in G_1_ population (60.4%) of SCC90 cells with a simultaneous decrease in G_2_/M phase (18.5%) in a dose-dependent manner distribution (Figure 3A,C). Thus, a significant increase in the cell percentage in the S phase and a decrease in the G_2_/M (76.1% to 22.3%, *p* < 0.05) were observed at 10 μg/mL. Remarkably, at 100 µg/mL, cells increased in the S phase compared to the *C. sinensis* (63.6% vs. 20.3%), and the G_2_/M distribution decreased (Figure 3A,C). Conversely, *L. barbarum* extract induced a decrease in the G_2_/M phase in CAL27 cells with all three concentrations and a non-significant increase in G_0_/G_1_ at 100 μg/mL (Figure 3B,D). The *C. sinensis* extract at 100 μg/mL produced effects in the SCC090 line, explicitly increasing cellular accumulation in the G_0_/G_1_ phase, reducing the distribution values in the S phase, and avoiding cell arrest in G_2_/M (*p* < 0.05) (Figure 3C), whereas no change in cell distribution of any phases for CAL27 cells in comparison to the negative control (cells not treated) were evidenced (Figure 3D).

### 2.4. L. barbarum Induces Downregulation of Viral Oncoproteins E6 and E7 and Enhances p53 Levels in SCC090 Cell Line

Considering that *L. barbarum* exhibited antiproliferative selectivity and induced changes in the SCC090-HPV16 cell cycle, we investigated the expression of the viral proteins E6 and E7 in treated and untreated cells. *L. barbarum* significantly reduced the mRNA expression of the E6 and E7 oncoproteins at 1.0 and 100 µg/mL, and the E6 protein expression decreased, involving a change of 0.22 (*p* ≤ 0.001) and 0.1 times at 10 µg/mL. Additionally, there was a significant decrease in mRNA of the E7 oncoprotein at test concentrations (i.e., 1.0, 10, and 100 µg/mL, *p* < 0.05) compared with the control extract *C. sinensis*, untreated cells, and cisplatin (20 µM) (Figure 4A).

Additionally, there was a 9.22-fold increase at 1.0 µg/mL (*p* < 0.05) and a 2.1-fold increase at 100 µg/mL of the tumor-suppressor protein p53 gene expression in *L. barbarum*-treated SCC090 cells (*p* < 0.05) (Figure 4B). Therefore, the *L. barbarum* extract decreased the expression of the viral oncoproteins E6 and E7 and increased the levels of p53, enhancing its therapeutic potential against HPV16-positive OSCC cells to be further explored. Regarding the *C. sinensis* control extract, its effect on p53 expression behaves in a dose-dependent manner. Accordingly, in comparison to the negative control, they promoted a gene overexpression (*p* < 0.05), i.e., 8.5- and 23.3-fold increase, respectively, at 10 and 100 μg/mL doses. Cisplatin showed a lower increase in E6, E7 and p53 expression (Figure 4A,B).

### 2.5. L. barbarum Alters the Expression of Cell cycle Regulatory Proteins

We investigated the mechanism of G_0_/G_1_ and S phase arrest in the SCC090 cell line with *L. barbarum* treatment by evaluating the expression levels through immunohistochemistry (IHC) of four G_0_/G_1_ and S checkpoint proteins, namely p53, p16, Bcl-2, and Ki-67. The expression of the cell cycle-related proteins was evidenced in cells showing nuclear and cytoplasmic stains (Figure 5).

There was an increase in the expression and nuclear accumulation of p53, and its downstream effector p16, after treating cells with 1.0 μg/mL *L. barbarum* (Figure 5(A1)), showing strong cytoplasmic and nuclear stain (Allred score = 8). The 20 µM cisplatin treatment increased p16 expression, with an Allred score of 7 (Figure 5(A5)).

The control extract of *C. sinensis* at 1.0 μg/mL showed less than 10% of p53 nuclear and cytoplasmic stain (Allred a score = 2) (Figure 5(B3)) and showed moderate intensity at 10 μg/mL (Allred score = 4) (Figure 5(B4)). Although there was a decrease in the number and size, the chemotherapeutic drug cisplatin (20 μM) did not induce p53 protein expression in these cells (Figure 5(B5)).

Immunohistochemical staining for Ki67 showed that 1.0 μg/mL *L. barbarum* treatment reduced the number of actively proliferating tumor cells (Allred score = 2) compared with untreated control cells (Figure 5(C1)). In contrast, at 10 μg/mL, 20% of cells exhibited nuclear localization of staining, with an intermediate Allred score of 4 (Figure 5(C2)). *C. sinensis* exhibited no differences from the control cells, with less than 15% nuclear expression (Allred score = 2), but the cell morphology changed (Figure 5(C3,C4)). Bcl-2 protein expression was absent in both control cells and cells treated with *L. barbarum*.

## 3. Discussion

Oral and mostly oropharyngeal HPV-positive tumors have unique clinic-pathological, molecular, and prognostic features that occur in young adults and are not associated with exposure to tobacco and alcohol [33], being the HPV16 the most common genotype [34,35]. In the present study, we further explored the antitumor-based biological activity of *L. barbarum* extract on human OSCC cell line expressing high-risk HPV16.

As aforementioned, phenolics are the typical chemical constituents of several plant parts of *L. barbarum*, especially the fruit [20,36], and several of such metabolites have been previously isolated [21]. According to the LC-ESI-HRMS analysis of *L. barbarum* extract, 100% of the compounds were related to phenolics, including nitrogenous phenolics, single phenylpropanoids, anthocyanins, flavan-3-ols, flavonols, and proanthocyanins. The last two flavonoid-like metabolites exhibit important properties that benefit human health, especially in the context of cardiovascular health [37], metabolic disorders [38], antioxidant action [39,40], and cancer prevention [39], and have been recognized as dietary compounds. In the test extract of *L. barbarum* fruits, the chemical composition was more related to flavonoids (i.e., 58.65% relative abundance), and the remaining content included nitrogenous phenolics and single phenylpropanoids (Table 1). These phenolics covered the TPC and TFC and coincided with the relative abundance of detected/identified flavonoids, which was higher than reported elsewhere for extracts of *L. barbarum* fruits [41].

The most abundant identified metabolites in the test *L. barbarum* extract were 3′,7-*O*-dimethylcatechin 3-*O*-glucoside, 5-*O*-caffeoylshikimic acid, and *N*-feruloyl tyramine (34.87%, 21.03%, and 12.02% relative abundance, respectively), whose structurally related compounds exhibited a wide range of activities. There is evidence that some plant sources containing many flavonols and flavan-3-ols (such as quercetin and catechin-like flavonoids, respectively) play essential roles in cancer chemoprevention [42]. The activity of these naturally occurring compounds has been explained by their high antioxidant capacity, as demonstrated previously in several in vitro experiments [43,44]. However, the number of free hydroxyl groups for these flavonoids is crucial, so the transformation to methylated or glycosylated derivatives can modify the biological activity, improving or reducing the beneficial effect by making it easier or more challenging enter and/or accumulate in the target cell [45].

A previous study also demonstrated that quercetin derivatives were found to be more active against Caco-2, MCF-7, and BxPC-3 tumor cell lines than catechin derivatives [46]. However, gallocatechin-like compounds have displayed activity against MCF-7, HT-29, and UACC-375 cancer cell lines and even in vivo reduction in tumor incidence [47]. In the case of 5-*O*-caffeoylshikimic acid, it exhibited moderate multidrug resistance reversal activity on a human mdr1 gene-transfected mouse lymphoma cell line [48]. The other minor compounds, corresponding to derivatives of single phenylpropanoids and flavonoids, have exhibited antioxidant effects, determined by the 2,2-diphenyl-1-picrylhydrazyl (DPPH) radical scavenging and the β-carotene decoloration [44,49].

Previous studies have indicated that *N*-coumaroyl and *N*-feruloyl tyramines exhibited free radical scavenging and inhibitory effects on lipid peroxidation in rat liver microsomes [29] and nitric oxide (NO) production [50]. A set of six tyramine- and dopamine-conjugated HCAAs, involving caffeoyl, coumaroyl, and feruloyl moieties, were previously evaluated against OSCC cell lines [51]. The results indicated that the *N*-caffeoyl derivative conjugated with tyramine showed relatively higher cytotoxic selectivity toward OSCC cell lines (IC_50_ = 30 µM), whereas the *N*-feruloyl derivative exhibited lower cytotoxicity (IC_50_ = 192). Remarkably, dopamine appeared to have a relevant impact on the cytotoxicity against OSCC cell lines, regardless of the acid residue, since the activity was very similar for the three hydroxycinnamic acid dopamine derivatives (44 µM > IC_50_ > 51 µM). In our study, *N*-feruloyl tyramine was the most abundant HCAA (12.02%) in the investigated *L. barbarum*-derived extract, whereas *N*-caffeoyl tyramine and *N*-caffeoyl dopamine had lower abundance (0.69 and 2.04%, respectively). However, HCAAs from *L. barbarum* have displayed immunomodulatory activity, especially *N*-feruloyl 3-*O*-methyldopamine, and even the lignamide grossamide [24], both detected in the studied extracts at low abundances (1.25 and 2.83%, respectively). HCAAs and lignamides are considered active principles in *Lycium*-derived extracts [30]. Thus, amides could contribute directly or indirectly to the activity against OSCC cell lines of *L. barbarum* extract, but the most active compounds seem to be related to the minor HCAA components, which would explain the observed low antiproliferative activity of the extract. Consequently, the effect of this test *L. barbarum* extract can also be explained by the combined action of various phenolics and flavonoids, mostly catechin- and quercetin-related flavonoids and tyramine- and dopamine-related HCAAs. Several studies suggest that phenolic mixtures can affect multiple targets by additive synergism against cancer progression and development, although some antagonistic events can occur. Therefore, such combinations might be considered safe and valuable approaches for cancer therapy and prevention [52,53] if a detailed analysis is performed, which deserves exploration in future studies.

Considering the cytotoxicity criteria of the National Cancer Institute (NCI, USA) for natural products, a treatment is considered moderately cytotoxic if the IC_50_ is between 10 and 20 μg/mL, cytotoxic < 50 μg/mL, and highly cytotoxic < 10 μg/mL [54]. In this regard, *L. barbarum* showed modest cytotoxic activity (IC_50_ > 100 μg/mL) on SCC090 and CAL27 at 48 h, whereas it was safe for healthy HGnF. Previous studies have showed that *L. barbarum* induced apoptosis of T47D breast cancer cells at 1.0 mg/mL by increasing the expression of the pro-apoptotic protein Bax [55]. Furthermore, it induced apoptosis in human prostate cancer cells PC-3 and DU-145 at higher concentrations (400–1000 µg/mL) through the regulation of the expression of Bcl2/Bax proteins [56], inhibiting proliferation and apoptosis of human cervical carcinoma (HeLa) cells via the mitochondrial pathway [57]. On the other hand, Gong et al. demonstrated that the arabinogalactan fraction obtained from *L. barbarum* could arrest the MCF-7 cell cycle in the G_0_/G_1_ phase; significantly regulate the expression of Bax, Bcl-2, and Caspase-3, 8 and 9; decrease mitochondrial membrane potential; increase the ROS production; and alter the expression of phosphorylated Erk, JNK, and p38 proteins [58]. Chen et al. observed the cytotoxic effects of *L. barbarum* polysaccharides on a DU145 prostate cancer cell line, reporting an IC_50_ of 374.11 μg/mL [59]. Although our results obtained herein for SCC090 and CAL27 suggest that the extract of *L. barbarum* exhibits modest antiproliferative action at the time and investigated doses, the observed effects agree with those reported for tumor cell lines from other tissues.

The growth and proliferation of HPV-positive cancer cells depend on the expression of the viral E6/E7 oncogenes and inactivation by the degradation of tumor-suppressor p53. In this regard, the inhibition of the expression of E6 and E7 could also provide an attractive therapeutic strategy [60,61]. We found that *L. barbarum* exhibited its antiproliferative activity in the SCC090 HPV16 cell line by inducing G_1_/S phase arrest by increasing mRNA expression and protein of p53 levels at 1.0 μg/mL. Hence, the p53 upregulation by *L. barbarum* might result in p53-dependent G_1_/S arrest. Previous studies have demonstrated that *L. barbarum* inhibits growth and induces G_1_/S and G_2_/M arrest in several cancer cells [62]. In HeLa cells, HPV18-positive *L. barbarum* regulates the growth by arresting the cells at the S phase [63] through transcriptional activation of Mdm2 and TERT genes, which induce p53 protein expression [64].

The viral E7 oncogene of HPV-16 is known to inactivate the complex formation between pRb and E2F; this alteration leads to deregulation of the cell cycle [65,66,67]. We observed that *L. barbarum* downregulates the expression levels of E6 and E7 genes at 1.0 μg/mL, which might have caused the eventual arrest of cells in the G_1_/S phase and restoration of p53 and pRb proteins in SCC090 cells. In this sense, p53 activation could lead cells to either initiate apoptosis or undergo cell cycle arrest, as has been reported: low levels of p53 induce cell cycle arrest, and high levels induce apoptosis [68]. Several phytochemicals have been shown to have an inhibitory effect on E6/E7 oncogene transcription. Loss of E6 and E7 levels by curcumin and berberine is related to the downregulation of transcription factors AP-1 in cervical cancer cells [69,70,71]. Similarly, the catechins from *C. sinensis* have shown potent activity against HPV-positive carcinoma cells, inhibiting cell proliferation and HPV E6/E7 expression [39]. Conversely, it has been reported that 10–33 μM cisplatin can strongly reduce HPV16 and HPV18 E6/E7 oncogene expression in HPV-positive cancer cells [72]. However, 20 μM cisplatin concentration did not cause any inhibition of HPV E6/E7 expression in SCC090 cells.

Under normal conditions, p16 is a negative regulator of CDK4 and cyclin D, which prevents it from phosphorylating Rb and inducing cell cycle arrest [73,74]. The SCC090 cells used in this study overexpress the CDKN2A gene [75] and express high levels of the p16 protein [76]. Oncoprotein HPV16 E7 has been shown to trigger p16INK4A expression as a mechanism used by cells to stop the cell cycle [77]. Interestingly, we observed strong p16 immunostaining with *L. barbarum* treatment, which might have resulted in the observed G_1_ cell cycle arrest [78]. Additionally, the p53-dependent apoptosis induction [79] in response to the p16 overexpression has been observed in various cancer cell lines [80]. Moreover, our data show a decrease in ki67 protein, a marker for cell proliferation [81], in tongue cancer cells treated with the *L. barbarum* extract at 1.0 μg/mL. This finding suggests that this extract may target the expression of Ki67 in SCC090 cells and plays a role in preventing cell proliferation that could ultimately inhibit cancer progression.

## 4. Materials and Methods

### 4.1. Botanical Extracts

Ethanolic dry extract of Chinese *L. barbarum* was obtained from commercially-purchased fruit powder (Greena Health center, Xi’an, China). *C. sinensis* (GOSLIM^®^, INVIMA registration: NSA-001154-2016) leaves were used as a positive control.

### 4.2. Chemical Characterization of L. barbarum Extract

#### 4.2.1. Measurements of Total Phenolic and Total Flavonoid Contents

Total phenolic content (TPC) using Folin-Ciocâlteu (FC) reagent and total flavonoid content (TFC) using aluminum chloride (AlCl_3_) were measured by colorimetry [82]. For TPC, a mixture comprising extract (5 mg/mL, 20 μL), 10% FC reagent (100 µL), and 7.5% Na_2_CO_3_ (80 µL) was prepared and incubated for 1 h at room temperature in the dark. Absorbance was recorded at 760 nm using a VarioskanTM LUX 96-well plate reader (Thermo Fisher Scientific, Waltham, MA, USA). TPC was expressed as milligrams of gallic acid equivalents (GAE) per gram extract dry weight (mg GAE/g Edw). For TFC, a mixture comprising extract (5 mg/mL, 100 μL), 10% AlCl_3_ (30 µL), 0.1 M CH_3_COONa (30 µL), and ethanol (80 µL) was prepared and incubated for 30 min at room temperature in the dark. Absorbance was recorded at 420 nm. TFC was expressed as mg catechin equivalents (CE) per gram Edw (mg QE/g Edw).

#### 4.2.2. Liquid Chromatography-Mass Spectrometry Analysis

*L. barbarum* dry crude extract was resuspended in absolute ethanol and analyzed through liquid chromatography (Prominence^TM^, Shimadzu, Columbia, MD, USA) coupled with a high-resolution mass spectrometer (micrOTOFQII™, Bruker, Billerica, MA, USA) with an electrospray ionization source (ESI) operated in positive ion mode ([M + H]^+^). Chromatographic separations were performed using a Phenomenex Luna^®^ (Phenomenex, Torrance, CA, USA) C18 HPLC column (150 mm × 2.0 mm, 3 μm) and a combination of 0.1% (*v*/*v*) formic acid in H_2_O (solvent A) and LCMS-grade acetonitrile (ACN) (solvent B) at 1.0 mL/min. An optimized gradient elution method was employed: (0–2 min 10% B, 2–4 min 10% to 30% B, 4–5 min 30% B, 5–10 min 30% to 100% B, 10–13 min 100% B, 13–16 min 100% to 0% B, and 16–20 min 10% B). UV detection was performed at 270 nm. ESI was operated in positive ion mode (scan 100–1500 *m*/*z*), desolvation line temperature of 250 °C, nitrogen as nebulizer gas at 1.5 L/min, drying gas at 8 L/min, quadrupole energy of 7.0 eV, and collision energy of 14 eV. Total ion chromatograms (TIC) were processed in MzMine 2 software (Whitehead Institute for Biomedical Research, Cambridge, MA, USA). Compounds were annotated after detailed scrutiny of the ultraviolet-visible and high-resolution mass spectral (exact mass of quasi-molecular and fragment ions) data of each signal, compared to the information previously reported in the literature for *Lycium* species [25,26,27,30].

### 4.3. In Vitro Culture of SCC090 Cells and CAL 27 Cells

The UPCI: SCC090 (CRL-3239™, ATCC^®^ (American type culture collection), Manassas, VA, USA) cell line from a squamous cell carcinoma of the base of the tongue containing human papillomavirus 16 (HPV16) DNA sequences, CAL 27 (ATCC^®^ CRL-2095™) derived from the tongue with squamous cell carcinoma cancer negative for HPV, and normal human gingival fibroblasts (HGnF, Catalog #2620, Sciencell, Carlsbad, CA, USA) were used in this study. The cells were cultured and routinely maintained in Dulbecco’s modified Eagle’s medium supplemented with fetal bovine serum 10% (Lonza) and penicillin–streptomycin–amphotericin B solution 1% (Lonza) in a humidified atmosphere of 5% CO_2_ at 37 °C [83,84,85].

### 4.4. Antiproliferative Activity Assay

The effect of extracts on the growth and proliferation of tumor and normal cells was determined using Alamar blue assay (Biosource, Camarillo, CA, USA) [86]. The cell lines were seeded in 96-well plates at a density of 1.0 × 10^4^ cells/well attachment allowed for 24 h. Stock solutions of the extracts (10 mg/mL) were prepared in 10% dimethyl sulfoxide (Sigma-Aldrich catalog #Q3251) and serially diluted (0–2000 µg/mL) and added to the 96-well plate, the cells being treated for 24 h and 48 h with extracts dilutions. At the end of the treatment period, the culture medium was replaced by 100 μL of resazurin solution (0.44 μM), and the plates were incubated for 4 h. After incubation, the fluorescence produced by resofurin was measured using a microtiter plate reader at 530–590 nm (Tecan, Infinite^®^ 200 PRO, Männedorf, Switzerland). The unstimulated samples were the survival control, and the chemotherapeutic drug cisplatin (Alpharma) at 20 μM was used as a standard.

Cell viability was expressed as a percentage of live cells relative to untreated controls. Dose–response curves of the percentage of cell viability plotted against concentrations of the extracts were constructed. IC_50_ was calculated from a linear regression between viability cells, and the concentration of samples [87] was determined using GraphPad Prism 7.0 software (GraphPad Corp., San Diego, CA, USA). Each experiment was performed as three independent tests, with a coefficient of variation of less than 20%.

### 4.5. Selectivity Index Analysis

The selectivity index (SI) was calculated from the IC_50_ of a compound against normal cells (HGnF) divided by the IC_50_ value of cancer cells (SCC090, CAL27) [87,88]. Extracts are classified as high selectivity if the SI value is >1 and less selective if the SI value is <1 [88] as calculated from Equation (1).
(1)$$\mathrm{SI}=\frac{{\mathrm{IC}}_{50}\;\mathrm{normal}\;\mathrm{cells}}{{\mathrm{IC}}_{50}\;\mathrm{cancer}\;\mathrm{cells}}$$

### 4.6. Cell Cycle Analysis

Cells lines (2 × 10^4^ cells) were plated in 24-well cell culture plates, with attachment allowed for 24 h. Cells were treated with *L. barbarum* and *C. sinensis* 1.0, 10 and 100 μg/mL by 48 h; cells untreated and treated with cisplatin 20 μM were included as a control. At the end of each treatment time, cells were collected by trypsinization and washed twice with PBS, and a cell pellet obtained was resuspended in 200 μL fixed and permeabilized with 80% ethanol for 2 h. The fixed cells were then washed with PBS and stained with 7-Amino Actinomycin D (7-AAD) solution 25 mg/mL (BD Pharmingen™, San Diego, CA, USA) for 15 min at room temperature in the dark. Stained cells were analyzed for DNA content using a BD AccuriTM C6 flow cytometer (BD Biosciences, San Jose, CA, USA), and the percentages of cells in G0/G1 phase, S phase, and G2/M phase were established in a trial version of the ModFit LT software, version 5.0. All experiments involved triplicates.

### 4.7. RNA Extraction and RT-PCR Analysis

SCC90 cells (5 × 10^5^ cells) were plated in T25 cell culture flasks, allowing the attachment for 24 h. Cells were treated with *L. barbarum* and *C. sinensis* 1.0, 10, and 100 μg/mL by 24 h, and cells untreated and treated with cisplatin 20 μM were included as a control. Total RNA was extracted from cells using Quick-RNA MicroPrep kit (Zymo Research, Irvine, CA, USA). RNA quality was evaluated using a Nanodrop (Thermo Scientific, Waltham, MA, USA) to calculate the concentration and to assess the purity.

Real-time RT-PCR was performed using real-time Luna^®^ Universal One-Step RT-qPCR master mix reagent (New England Biolabs, Ipswich, MA, USA) and CFX96 Real-Time PCR Detection Systems (Bio-Rad Laboratories, Hercules, CA, USA). Primers were designed using Beacon Designer software and are listed in Table 3.

The reaction mixture comprised 4.1 µL of the template (10 ng/µL), 5 µL of Luna Universal One-Step Reaction Mix [2x], 0.5 µL of Luna WarmStart RT Enzyme Mix [20X], and 0.2 µL of primers [10 µM] in a final reaction volume of 10 µL. The PCR amplification consisted of an initial retrotranscription step at 55 °C for 10 min, followed by a denaturation step at 95 °C for 10 min, and 40 cycles of PCR at 95 °C for 15 s and at 60 °C for 60 s. Expression levels were calculated from the qPCR results based on the modified 2^-ΔΔCt^ method suggested by Pfaffl [89]. For these calculations, GAPDH and unstimulated cells were used as controls.

### 4.8. Immunohistochemistry

SCC090 cells (2 × 10^4^ cells) were plated in four-well chamber slides (Nunc™ Lab-Tek™ II Chamber Slide™ System, ThermoFisher Scientific, Waltham, MA, USA) for 24 h. Cells were treated with L. *barbarum* and *C. sinensis* at 1.0 and 10 μg/mL and cisplatin (20 μM) for 48 h. Cells were then washed twice with PBS and fixed with 4% paraformaldehyde. (Sigma-Aldrich, Burlington, MA, USA). Immunohistochemistry (IHC) was performed to study the expression levels of Ki67 (Dako clone MIB-1), p53 (Dako Clone DO7), Bcl-2 (Dako clone 124), and p16 in an autoimmunostainer (Ventana XT System Benchmark; Ventana Medical Systems, Oro Valley, AZ, USA) according to the manufacturer’s recommendations. Two pathologists evaluated the protein expression levels. The typical areas for each case were photographed with a 20× objective lens using a Zeiss microscope and digital camera (Zeiss Axio Imager A2, Zeiss Microscopy GmbH, Jena, Germany).

The Allred method determined the intensity score and proportion of marked cells, which combines the percentage of positive cells and marking intensity [90]. The Allred total score was calculated from Equation (2), where the ISR = intensity score ratio, PS = percentage score, IS = intensity score, and IS_0–8_ = intensity score ratio were within the 0–8 range.
(2)$${\mathrm{Allred}\;\mathrm{total}\;\mathrm{score}=\mathrm{ISR}+\mathrm{PS}+\mathrm{IS}}_{0–8}+\mathrm{IS}\;$$

### 4.9. Data Analysis

All experiments were carried out in triplicate to obtain comparative data. Parametric data were analyzed using a two-way ANOVA, and nonparametric data with the Kruskal–Wallis test using GraphPad Prism 6 software. Results were expressed as means ± the standard error of the mean (SEM), obtained from three independent experiments; *p*-values < 0.05 were considered statistically significant for all tests.

## 5. Conclusions

The present study is the first evidence that *L. barbarum* extract has a potential role in preventing cancer progression by inhibiting the growth and proliferation of head neck cancer cells through the induction of the G_1_/S and G_2_/M arrest, and the upregulation of tumor-suppressor proteins p53 and p16, which is most likely a consequence of the reduced levels of E6 and E7, respectively. Although the antiproliferative effect was found to be modest (IC_50_ < 100 µg/mL) at 48 h, the observed effects of this phenolic-rich extract from *L. barbarum* fruits can be rationalized by the individual or combined action of flavonoids and/or HCAAs, especially the catechins and tyramine/dopamine amides conjugated with an *N*-caffeoyl residue. These facts require additional studies to define optimum exposition times and the respective roles of active principles of the studied extract. In this sense, our finding further suggests that the therapeutic potential of this extract in HNSCC can be examined deeper in future studies.

## Figures and Tables

**Figure 1 molecules-27-03568-f001:**
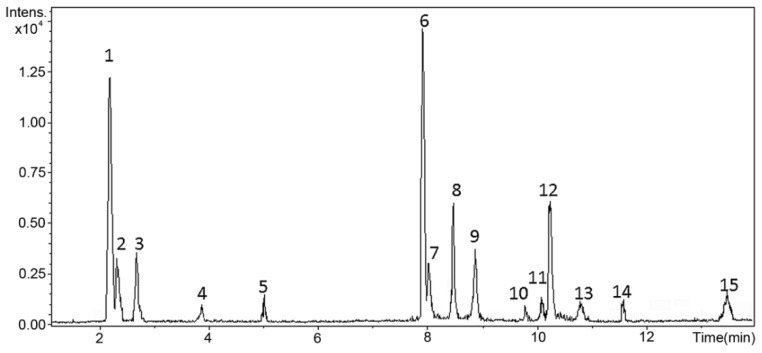
Total ion chromatogram (TIC) of the ethanol extract from *L. barbarum* fruits.

**Figure 2 molecules-27-03568-f002:**
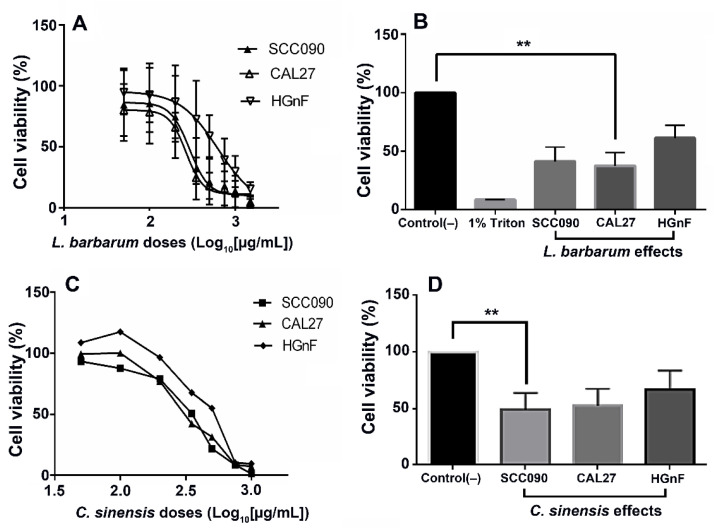
Cytotoxic activity of extracts from *L. barbarum* and *C. sinensis* on tumor cell lines SCC090, CAL27, and GHnF. Cells were treated for 48 h at different concentrations (50–1000 μg/mL). (**A**) IC_50_ curve of the effect of *L. barbarum*. (**B**) Comparison of the IC_50_ of *L. barbarum* in each cell line. (**C**) IC_50_ curve of *C. sinensis*. (**D**) Comparison of the IC_50_ of *C. sinensis* in each cell line. The results are expressed as the mean of three independent tests with their three replications (*n* = 3) ± standard error of the mean (SEM). Two asterisks (**) represent statistically significant differences in the extract effect regarding the control treatment (untreated cells), comparing the effect of the extract on cell lines (*p* < 0.05).

**Figure 3 molecules-27-03568-f003:**
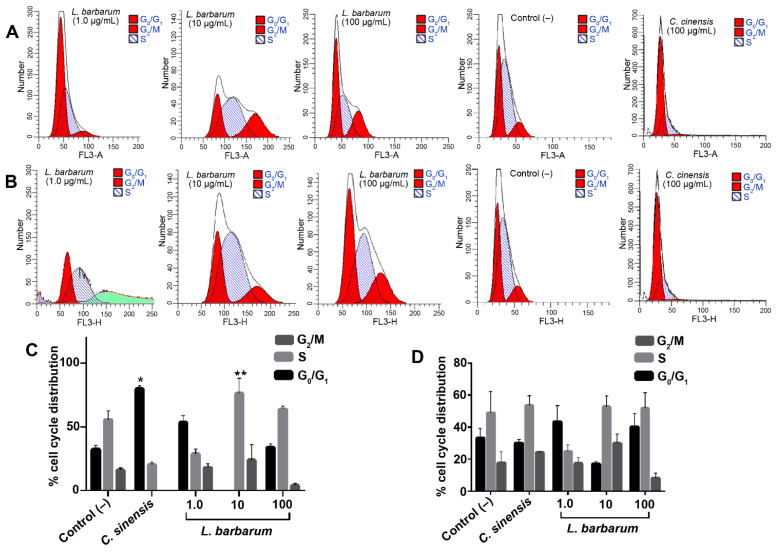
*L. barbarum* extracts induced cell cycle arrest in SCC090 and CAL27 cells at the concentrations evaluated. Histograms for SCC090 (**A**) and CAL27 (**B**) cells in different phases of the cell cycle analyzed by the ModFit software are shown. Quantification of a cell population (in percentage) in different phases of the cell cycle is shown as bar diagrams for SCC090 (**C**) and CAL27 (**D**) cells. Results are expressed as the mean of three independent trials in triplicate (*n* = 3) ± standard error of the mean (SEM). The effect of the extract concerning the treatments was compared: * control (–) (untreated cells) and ** control (+) (*C. sinensis* extract at 100 μg/mL) (*p* < 0.05).

**Figure 4 molecules-27-03568-f004:**
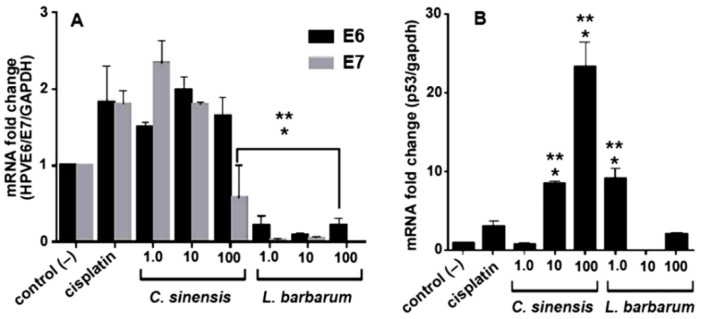
mRNA expression levels in SCC090 cells treated with extracts from *L. barbarum* and *C. sinensis* for 48 h at 1.0, 10, and 100 μg/mL. The mRNA levels were measured by reverse transcription-quantitative polymerase chain reaction (RT-qPCR): (**A**) E6 and E7, (**B**) p53. Results are expressed as the mean of three independent triplicate trials (*n* = 3) ± standard error of the mean (SEM). The effect of the extract was compared concerning the treatments: * control (–) (untreated cells) and ** control (+) cisplatin (*p* < 0.05).

**Figure 5 molecules-27-03568-f005:**
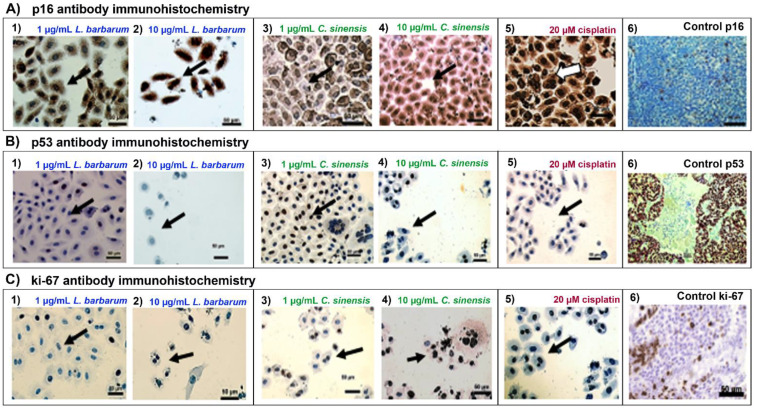
Immunohistochemical images for SCC090 cell line on evaluating the expression levels of G_0_/G_1_ and S checkpoint proteins using (**A**) p16, (**B**) p53, and (**C**) Ki-67 antibodies, after treating with (**1**) 1.0 μg/mL *L. barbarum*, (**2**) 10 μg/mL *L. barbarum*, (**3**) 1.0 μg/mL *C. sinensis*, (**4**) 10 μg/mL *C. sinensis*, (**5**) 20 μM cisplatin, and (**6**) negative control (no treatment). (Scale Bar 50 μm). Observations for the (**A–C**) × **1–6** pictures: (**A1**): 70% nuclear and cytoplasmic expression; (**A2**): intense staining and change in cell size and shape; (**A3**): more than 70% nuclear and cytoplasmic staining; (**A4**): slight change in shape and size with intense staining; (**A5**): more than 70% nuclear and cytoplasmic staining; (**A6**): cervix, showing intense staining; (**B1**): intermediate expression; (**B2**): no expression, but there was a change in the shape and size of SCC090 cells; (**B3**): nuclear single intermediate expression; (**B4**): nuclear weak expression with cell size and shape alterations; (**B5**): absent staining with a slight change in cell shape; (**B6**): gastric, showing intense staining of more than 70% nuclear and cytoplasmic; (**C1**): weak expression; (**C2**): intermediate expression and a slight change in cell morphology; (**C3**): weak expression; (**C4**): weak expression, but the cells displayed morphological changes; (**C5**): weak expression; (**C6**): amygdala, showing intermediate staining.

**Table 1 molecules-27-03568-t001:** Main chemical constituents of *L. barbarum* extract annotated by liquid chromatography coupled with high-resolution mass spectrometry with elec-trospray ionization (LC-ESI-HRMS).

#	rt ^a^	RA ^b^	*m*/*z*^c^	Molecular Formula ^d^	Error (ppm) ^e^	Fragment Ions ^f^	Name ^g^
	(min)	(%)	([M + H]^+^)				
**1**	2.22	21.03	337.0938	C_16_H_17_O_8_	4.45	175.0621	5-*O*-caffeoylshikimic acid
**2**	2.37	3.57	607.1795	C_32_H_31_O_12_	−3.46	305.1018, 125.0611	4′,4′′′-*O*-dimethylprocyanidin B
**3**	2.7	4.03	291.0854	C_15_H_15_O_6_	−5.02	182.0579	catechin
**4**	3.9	0.64	331.0805	C_17_H_15_O_7_	−3.87	195.0293	3′,4′-*O*-dimethylquercetin
**5**	5.03	0.81	301.0721	C_14_H_13_O_6_	−2.99	192.0433	7-methoxyluteolin
**6**	7.92	34.87	481.1702	C_23_H_29_O_11_	−1.66	319.1172, 197.0802	4′,7-*O*-dimethylcatechin 3-*O*-glucoside
**7**	8.03	3.85	463.1228	C_22_H_23_O_11_	−2.59	301.0718, 179.0351	4′-*O*-methylcyanidin 3-*O*-glucoside
**8**	8.48	6.36	611.1621	C_27_H_31_O_16_	1.46	449.1072, 303.0511, 311.1331	quercetin 3-*O*-rutinoside (rutin)
**9**	8.91	4.52	481.1723	C_23_H_29_O_11_	2.70	319.1163, 197.0806	4′,5-*O*-dimethylcatechin 3-*O*-glucoside
**10**	9.79	0.69	300.1246	C_17_H_18_NO_4_	−3.33	163.0384, 137.0855	*N*-caffeoyl tyramine
**11**	10.22	1.45	284.1279	C_17_H_18_NO_3_	−2.82	147.0455, 137.0851	*N*-coumaroyl tyramine
**12**	10.26	12.02	314.1399	C_18_H_20_NO_4_	−2.23	177.0563, 137.0853	*N*-feruloyl tyramine
**13**	10.81	2.04	316.1195	C_17_H_18_NO_5_	−3.16	177.0559, 153.0779	*N*-caffeoyl dopamine
**14**	11.58	1.25	344.1486	C_19_H_22_NO_5_	3.49	163.0387, 167.0958	*N*-feruloyl 3-*O*-methyldopamine
**15**	13.47	2.87	625.2539	C_36_H_37_N_2_O_8_	1.76	489.1776, 352.0958, 137.0861	grossamide

^a^ rt = retention time; ^b^ RA = relative abundance from normalized peak intensity in total ion current chromatograms; ^c^ accurate mass of the quasi-molecular ion ([M + H]^+^); ^d^ Molecular formula deduced from accurate mass measurements; ^e^ calculated from the respective monoisotopic mass; ^f^ fragment ions observed in MS/MS spectra from [M + H]^+^ as precursor ion; ^g^ Identified compounds on the basis of diagnostic evidence, phylogenetic filtering and database and literature data comparison.

**Table 2 molecules-27-03568-t002:** Selectivity index and IC_50_ of *L. barbarum* and *C. sinensis* extracts and cisplatin on SCC090, CAL27, and HGnF cell lines.

	SCC090 ^a^		CAL27 ^a^		HGnF ^a^
Treatments	IC_50_ ^b^	SI ^c^	IC_50_ ^a^	SI ^c^	IC_50_ ^a^
*L. barbarum* Extract	454.6	1.4	266.2	2.4	626.5
*C. sinensis* Extract	371.6	1.3	316.2	1.5	471.6
Cisplatin	5.84	-	-	-	-

^a^ Cell lines: SCC090 (human papillomavirus 16 (HPV16)-positive squamous cell carcinoma (SCC) from tongue), CAL27 (HPV16-negative SCC from tongue), and HGnF (normal human gingival fibroblasts); ^b^ IC_50_ = Half-maximal inhibitory concentration expressed in µg/mL; ^c^ SI = selectivity index calculated using Equation (1).

**Table 3 molecules-27-03568-t003:** Sequences of primers (F and R) used for real-time PCR, sequence ID, and lengths of amplification products.

**Primer**	**Sequence (5′ to 3′)**	**Genbank** **Sequence ID**	**Product Size** **(bp)**
HPV 16 E6-F	GCACCAAAAGAGAACTGCAATGTT	MN705373.1	
HPV 16 E6-R	AGTCATATACCTCACGTCGCAGTA		152
HPV 16 E7-F	CAAGTGTGACTCTACGCTTCGG	MK343362.1	
HPV 16 E7-R	GTGGCCCATTAACAGGTCTTCCAA		81
p53-F	CAGCATCTTATCCGAGTG	MG595993.1	198
p53-R	CAGTGTGATGATGGTGAG		

## Data Availability

The authors confirm that the data supporting the findings of this study are available within the article and from the corresponding author upon request.

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
