# Peer review of "Antiproliferative and Pro-Apoptotic Effects of a Phenolic-Rich Extract from Lycium barbarum Fruits on Human Papillomavirus (HPV) 16-Positive Head Cancer Cell Lines"

_molecules, 2022, doi:10.3390/molecules27113568_

Round 1

Reviewer 1 Report

The manuscript is well organised and readable. 

I have only two suggestions:

1. Discussion section: there are a lot of repetitions from the results section and the introduction. I think that discussion should be changed and limited to discussing the results without repetition

2. Abbreviations should appear each time a word appears in the text for the first time. Many abbreviations are explained in the methods section

Author Response

Reply to reviewer 1.

Thank you for your kind review and suggestions for our manuscript. We have carefully addressed each of the comments and suggestions and generated accordingly a definitively improved version of the manuscript, highlighting in yellow the respective changes in the revised version. Concerning the raised comments, following we present our point-to-point responses:

Comments by Reviewer 1

Comment 1: The manuscript is well organised and readable.

Response: We kindly thank the reviewer for the positive comment and recognizing the information we presented in the manuscript. On addressing her/his concerns resulted in an improved manuscript version.

Comment 2: Discussion section: there are a lot of repetitions from the results section and the introduction. I think that discussion should be changed and limited to discussing the results without repetition.

Response: Thank you very much for your kind suggestion. After a careful scrutiny of the discussion section, we modified and removed some redundant passages with results and introduction as suggested.

Comment 3: Abbreviations should appear each time a word appears in the text for the first time. Many abbreviations are explained in the methods section

Response: We appreciate your kind remark. Accordingly, abbreviations were revised in detail and specified once they appeared for the first time in the manuscript. 

Reviewer 2 Report

The antiproliferative activity of a phenolic-rich extract from Lycium barbarum fruits against head and neck HPV16 squamous cell carcinoma (OSCC) was carried out and reported  L. barbarum extract inhibits human papillomavirus (HPV) type 16 cell lines. Studies showed that the observed effects were associated with anticancer and immunomodulatory  phenolics compounds  identified in this extract. It seems a very interesting study conducted by authors. 

Abstarct: Ok

Introduction:OK

Material & Methods:Page 393: what is the meaning of standardize extract here? are all the compounds mentioned in the studies already confirmed their presence in the ethanolic extract or something else? 

Results : As the authors mentioned, cytotoxicity criteria of the National Cancer Institute (NCI, USA) for natural products, a treatment is considered moderately cytotoxic if the IC50 is between 10 and 20 μg/mL, cytotoxic < 50 μg/mL, and highly cytotoxic < 10 μg/mL. Here authors have reported most susceptible (IC50 = 266.2 µg/mL- and least susceptible (IC50 = 626.5 µg/mL. In this case, how would you justify the potency of your extracts?

Fig2:Suggest authors should write different concentrarions as  (50-1000μg/mL).

Discussion :Well discussed

Table ;OK

Figure: Authors should increase the clarity of figure5.

Conclusion: Ok

Author Response

Reply to reviewer 2.

Thank you for your kind review and suggestions for our manuscript. We have carefully addressed each of the comments and suggestions and generated accordingly a definitively improved version of the manuscript, highlighting in yellow the respective changes in the revised version. Concerning the raised comments, following we present our point-to-point responses:

Comments by Reviewer 2

Comment 1: The antiproliferative activity of a phenolic-rich extract from Lycium barbarum fruits against head and neck HPV16 squamous cell carcinoma (OSCC) was carried out and reported L. barbarum extract inhibits human papillomavirus (HPV) type 16 cell lines. Studies showed that the observed effects were associated with anticancer and immunomodulatory phenolics compounds identified in this extract. It seems a very interesting study conducted by authors. 

Response: We kindly appreciate the positive comments by reviewer regarding our study and the information presented in the manuscript. The manuscript received an improvement on addressing her/his concerns.

Comment 2: Abstract: Ok; Introduction: OK; Table: OK; Conclusion: Ok; Discussion: Well discussed.

Response: Thanks a lot for your feedback. It is very important for us.

Comment 3: Material & Methods: Page 393: what is the meaning of standardize extract here? are all the compounds mentioned in the studies already confirmed their presence in the ethanolic extract or something else? 

Response: Thank you very much for your kind remark. The term “standardized extract” referred to a botanical extract that has components present in a specific amount, guaranteed by the extraction processing through batches. In this regard, we deduced such a standardization since all mentioned compounds have been informed and reported in previous studies as well as by the commercial availability and origin of this L. barbarum extract. However, the standardization was not confirmed and, therefore, to avoid any confusing terms, we decided to remove this term from this line of our manuscript.

Comment 4: Results: As the authors mentioned, cytotoxicity criteria of the National Cancer Institute (NCI, USA) for natural products, a treatment is considered moderately cytotoxic if the IC50 is between 10 and 20 μg/mL, cytotoxic < 50 μg/mL, and highly cytotoxic < 10 μg/mL. Here authors have reported most susceptible (IC50 = 266.2 µg/mL- and least susceptible (IC50 = 626.5 µg/mL. In this case, how would you justify the potency of your extracts?

Response: We appreciate so much your kind comment. During our studies, we observed that this phenolic-rich L. barbarum-derived extract exhibited low cytotoxic activity (IC50 > 100 μg/mL) as well as the Camelia sinensis extract, which indicated a low potency by direct antiproliferative effect. However, we observed that cancer cell lines exhibited a reasonable susceptibility than normal cell line and a particular behavior on inducing G1/S and G2/M arrest, upregulating the tumor-suppressor proteins p53 and p16, reducing the levels of E7/E5 oncogenes and the cell proliferation marker Ki67 at lower doses (< 10 μg/mL). Therefore, we justify the potency of the test L. barbarum extract related to the reduction of the cancer progression, as mentioned in manuscript’s conclusions.

Comment 5: Fig 2: Suggest authors should write different concentrations as (50-1000μg/mL).

Response: Thanks a lot for your kind suggestion. We modified the concentration range as suggested.

Comment 6: Figure: Authors should increase the clarity of figure 5.

Response: We appreciate so much your kind comment. Accordingly, we increased the clarity of Figure 5. First, we corrected a mistake related to the C. sinensis extract and, second, we reorganized the figure and comments in the figure caption to improve the clarity as suggested.

Round 2

Reviewer 1 Report

I accept the manuscript for publication in present form.